# Towards Building Affect sensitive Word Distributions

## Abstract

Learning word representations from large available corpora relies on the distributional hypothesis that words present in similar contexts tend to have similar meanings. Recent work has shown that word representations learnt in this manner lack sentiment information which, fortunately, can be leveraged using external knowledge. Our work addresses the question: *Can affect lexica improve the word representations learnt from a corpus ?* In this work, we propose techniques to incorporate affect lexica, which capture fine–grained information about a word's psycholinguistic and emotional orientation, into the training process of Word2Vec SkipGram, Word2Vec CBOW, and GloVe methods using a Joint Learning approach. We use affect scores from Warriner's affect lexicon to regularize the vector representations learnt from an unlabeled corpus. Our proposed method outperforms previously proposed methods on standard tasks for word similarity detection, outlier detection, and sentiment detection. We also show the usefulness of our approach for a new task related to the prediction of formality, frustration, and politeness in corporate communication.

## 1 Introduction

In natural language research, words, sentences and paragraphs are considered in context through vector space representations, rather than as atomic units with no relational information among them. Although n-gram based methods trained on large volumes of data have been found to outperform more complex approaches both on computational cost and accuracy, the techniques do not scale well in cases where the corpus size is limited(for example, for labeled speech or affect corpora with a size of a few millions of words). Recent work has attempted to improve the performance of word distributions for downstream tasks such as sentiment analysis (Sedoc et al., 2017) and knowledge base completion (Kumar & Araki, 2016) using lexical knowledge to enrich word embeddings, by performing methods such as regularization or introducing a loss term in the learning objective.

Sentiment relationships between words can be considered transitive, where 'good' < 'better' < 'best' implies that 'good' < 'best'. However, word representations based on traditional approaches such as Word2Vec (Mikolov et al., 2013) and GloVe (Pennington et al., 2014) are agnostic to the associated sentiments, emotions, or more generally affects (Bradley & Lang, 1999). Furthermore, although words such as *delighted* and *disappointed* share similar vector representations given their similar contexts, these words are associated with opposite reactions (or sentiments) as well as have a fairly different interpreted meaning. The challenge in using syntactic relational information for sentiment detection, is that sentiment relations are transitive and symmetric (i.e., if 'delighted' is the opposite of 'disappointed', then 'disappointed' is the opposite of 'delighted'.) Ignoring the bipolar nature of words could lead to spurious results, especially in predictive tasks related to synonyms and antonyms and sentiment analysis. On the other hand, incorporating affect–related information would make word distributions homogeneous and suitable for speech and text generation tasks that aim at capturing author or reader reactions. Furthermore, by using a small sentiment lexicon, it is possible to develop an automatic way to rate words based on their vector space representations. This could help reduce the time and cost required to gather word ratings, as well as eliminate the implicit biases that may be introduced in annotations, such as the high correlation between high valence ratings with high arousal reported by Sedoc et al. (2017).

We present an approach to build affect–enriched word representations. In other words, we enhance word distributions by incorporating reactions and affect dimensions. The output of this work produces word distributions that capture human reactions by modeling the affect information in the words. The affective word representations distinguish between semantically similar words that have varying affective interpretations. Affect is represented as a weighted relational information between two words, following the approach used by existing work. Sedoc et al. (2017) identify words of opposite polarity by performing signed spectral clustering on pre–trained embeddings. We present an approach to incorporate external affect and reaction signals in the pre–training step, using the hand-annotated affect lexica to learn from. Our experiments are based on using the state-of-the-art Warriner's affect lexicon (Warriner et al., 2013) as the input. The proposed approach builds on the intuition that relationships between synonyms and antonyms can be characterized using semantic dictionaries and the relationship can then be deterministically captured into the training loss functions.

We evaluate the proposed enriched word distributions on standard natural language tasks. We predict formality, frustration and politeness on a labeled dataset and show improved results using the enriched word embeddings. Further, we outperform the state–of–the–art for sentiment prediction on standard datasets. The key contributions of this paper include:

- Algorithm to incorporate affect sensors in the cost functions of distributional word representations (including Word2Vec SkipGram, Word2Vec CBOW, and GloVe) during training using semantic and external affect signals.

- Establish the utility of affect enriched word–embeddings for linguistic tasks such as Sentiment and Formality prediction in text data. Our method out performs the state–of–the–art with an 20% improvement in accuracy for the outlier detection methods. Detailed results are reported in table 1.

- Introduce a workflow to incorporate affective and reaction signals to word representations during pre–training. We show the generalizability of the workflow through experiments on 3 existing embeddings; Word2Vec–CBOW, Word2Vec–SkipGram, and GloVe.

Section 2 covers the prior art in both pre–training and post–training approaches for distributional word representations. Section 3 presents the proposed approach and detailed experiments are discussed in section 4. We conclude with a discussion on the learnings and the observations through this process 5.

## 2 RELATED WORK

A huge amount of research has explored how to use external resources to improve on Word2Vec (Mikolov et al., 2013) and GloVe (Pennington et al., 2014) embeddings. Research has refined embeddings for various downstream tasks such as dependency parsing (Bansal et al., 2014), sentiment analysis (Sedoc et al., 2017) and knowledge base completion (Kumar & Araki, 2016). There are mainly two approaches: joint learning and post-training. The former considers the incorporation of external knowledge into the training process of learning word embeddings itself. On the other hand, post-training approaches take already trained embeddings and use additional information to modify them. Our approach falls in the first category.

To the best of our knowledge, no prior work focuses on improving word embeddings by jointly learning from a corpus and an affect lexica. Bian et al. (2014) define a new basis for word representation and explore syntactic, semantic and morphological knowledge to provide additional information. A binary indicator function is used to define relations. Yu & Dredze (2014) propose a Relation Constrained Model(**RCM**) which predicts one word from another related word. They use a linear combination of the objectives in CBOW and RCM for joint learning.

Kiela et al. (2015) build specialized word embeddings for either similarity or relatedness. They use a joint learning approach by using additional context words from external sources with the SkipGram loss function. Our Word2Vec approach is similar to the pre-training model used in Kiela et al. (2015). Essentially, words in the pruned list $L^i_{pruned}$(section 3) can also be taken as additional context words, similar to their understanding. However, in our case, this addition is from an affect lexica and includes a strength associated with it. Xu et al. (2014) propose a general framework

**RC-NET** to incorporate knowledge into the word distributions. They encode external relational and categorical information into regularization functions and combine them with the original objective of Word2Vec SkipGram model. Our modified loss function can also be thought of as similar to the one in "Categorical Knowledge Powered model" in Xu et al. (2014), thinking of the strength as a similarity score, although the distance function there is just defined as the Euclidean distance. Bollegala et al. (2016) use a similar approach as ours for GloVe method to include various relations like synonyms, antonyms, hyponyms and so on. They make use of a binary function to indicate whether the relation between any two words exists or not.

For post-training, Faruqui et al. (2014) makes no assumption about input pre-trained word vectors. This work proposes an objective to further refine the input embeddings using relational information from semantic lexica. Another post training approach has been proposed in Mrkšić et al. (2016) which defines the final objective as a weighted sum of "Antonym Repel", "Synonym Attract" and "Vector Space Preservation" objectives.

## 3 JOINT LEARNING FROM UNLABELED CORPUS AND AFFECT LEXICA

### 3.1 NOTATIONS

Consider a corpus $C$ and an affect lexica $L$ consisting of $l$(word, affect-score) pairs. Let us denote the $i^{th}$ pair in $L$ with $p_i = (w_i, s_i)$. We define a function $S(i, j)$ which captures the strength of the relationship between any two words $w_i$ and $w_j$ in $L$.

### 3.2 OUR PROPOSED APPROACH

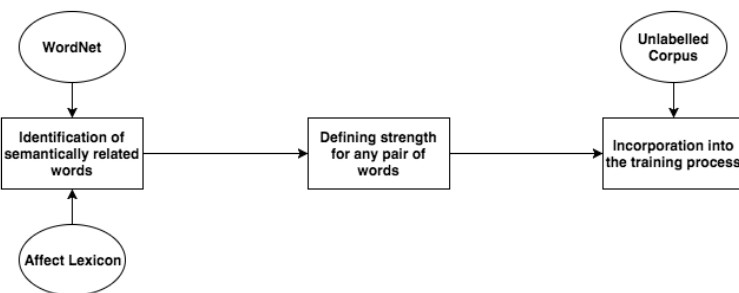

Figure 1: Workflow for the incorporation of affect information into the process of learning word embeddings.

To incorporate an affect lexica into the process of learning word embeddings, we follow a three step approach (See Figure 1):

**Step 1: Identifying Synonyms and Antonyms** - Given a pair $p_i$ in $L$, WordNet (Miller, 1995) is used to identify all pairs $p_j$ in $L$ such that $w_j$ is either a synonym or an antonym of $w_i$. Synonyms and antonyms are retrieved based on WordNet definitions. Note that, $S(i, j)$ will be defined as non–zero only when $w_j$ is returned by WordNet as the synonym or antonym of $w_i$. For example, consider a pair of words from the Warriner's affect lexicon (see section 4.1 for lexicon details): although 'confident' and 'funny' have similar valence scores of 7.56 and 7.59 respectively (on a scale of 1-9), they do not share a semantic relationship between them (i.e. not defined as synonyms or antonyms). Here, $S(i, j)$ is set to 0.

**Step 2: Defining strength** $S(i, j)$ **for all possible** $(i, j)$ **pairs** - Polarity information is captured in our modeling by centering the scores around 0 instead of the 1-9 scale. As already mentioned, $S(i, j)$ is defined as 0 if $w_j$ is not a synonym or antonym of $w_i$. For all other cases, the strength models the difference in affect scores of the two words under consideration. If the words have scores with the same sign, we define a positive strength inversely proportional to the relative distance between them. If the words have scores with opposite signs, the strength is negative, with magnitude directly proportional to the difference in their scores. Algorithm 1 describes the approach.

---

**Algorithm 1** Compute strength between two words

---

1: **function** GETSTRENGTH($s_i, s_j, smax, smin$) ▷ where $s_i$ and $s_j$ - normalized scores of $w_i$ and $w_j$, $smax$ and $smin$ - maximum and minimum normalized scores
2:
3:     strength = 0
4:     **if** $s_i s_j > 0$ **then**                                                ▷ Both scores have the same sign
5:         **if** $s_i > 0$ and $s_j > 0$ **then**
6:             strength = $|s_i - s_j|/smax$;                    ▷ $|.|$ refers to taking the absolute value.
7:         **else**
8:             strength = $|s_i - s_j|/smin$;
9:         **end if**
10:        strength = 1 - strength
11:    **else**
12:        strength = $-|s_i - s_j|/(smax\text{-}smin)$
13:    **end if**
14:
           **return** strength
15: **end function**

---

**Step 3: Loss function definition** - We introduce a new loss function for the embedding training. The loss functions defined for Word2Vec and GloVe are described here.

**Word2Vec loss function**: Rong (2014) describes the back–propagation algorithm of Word2Vec SkipGram and CBOW techniques. We build on top of that intuition. For the SkipGram approach, the model predicts the context of an input word in the corpus. Using the same notation as introduced in Rong (2014), the loss function of Word2Vec model (Mikolov et al., 2013) with negative sampling is defined as the following:

$$E_1 = -log\sigma(\mathbf{v}_{w_O}^{'T}\mathbf{h}) - \sum_{w_j \in W_{neg}} log\sigma(-\mathbf{v}_{w_j}^{'T}\mathbf{h}), \tag{1}$$

where $w_O$ is the output word (i.e. the positive sample) and $\mathbf{v}_{w_O}^{'}$ is it's output vector. $\mathbf{h}$ is the output value of the hidden layer. For the SkipGram approach, $\mathbf{h}$ is simply $\mathbf{v}_{w_I}$, which is the input vector corresponding to the input word $w_I$. $W_{neg} = \{w_j|1,...,K\}$ is the set of all negative samples. The standard unigram distribution raised to the $\frac{3}{4}$th power for this sampling is used for all reported experiments.

Information from affect lexica is incorporated using another loss function in the following manner:

$$E_2 = -\sum_{w_j \in L_{pruned}^i} S(i,j)log\sigma(\mathbf{v}_{w_j}^{'T}\mathbf{h}), \tag{2}$$

where $\mathbf{v}_{w_j}^{'}$ is the output vector as already defined, $\mathbf{h}$ is the hidden layer output, $i$ is the index of the input word $w_I$ in $L$ and $S(i,j)$ is the relation strength. $L_{pruned}^i$ is the pruned list of words obtained from WordNet corresponding to the input word $w_I$ (same as the word $w_i$ in $L$).

Note that $i$ will only be defined if the input word $w_I$ belongs to the affect lexica $L$. Hence, the loss function will only matter when a word present in $L$ appears at the input end.

The final loss function is defined as:

$$E = E_1 + \lambda E_2 \tag{3}$$

where $\lambda \geq 0$ is a hyper–parameter.

This modification results in a modified form of the derivative of $E$ with respect to $(\mathbf{v}_{w_j}^{'T}\mathbf{h})$, as given by the following equation:

$$\frac{\partial E}{\partial \mathbf{v}_{w_j}^{'T}\mathbf{h}} = \begin{cases} \sigma(\mathbf{v}_{w_j}^{'T}\mathbf{h}) - 1 & if \ w_j = w_O \\ \lambda S(i,j)(\sigma(\mathbf{v}_{w_j}^{'T}\mathbf{h}) - 1) & if \ w_j \in L_{pruned} \\ \sigma(\mathbf{v}_{w_j}^{'T}\mathbf{h}) & if \ w_j \in W_{neg} \end{cases} \tag{4}$$

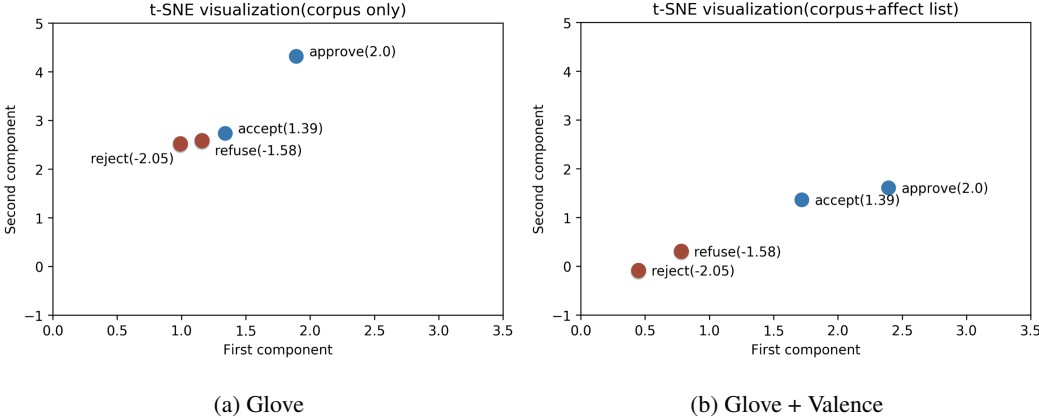

(a) Glove                              (b) Glove + Valence

Figure 2: t-distributed Stochastic Neighbor Embedding (t-SNE) visualizations of a subset of English words and their normalized valence scores. Incorporating valence information in word vectors separates the positive valence words from the negatives.

The weight update equations can be further obtained using equation 4 as in Rong (2014). Intuitively, if the strength is positive for a pair of words, the back–propagation algorithm makes their corresponding embeddings closer. Alternatively, if the strength is negative, the algorithm tries to move them apart. The magnitude of the strength controls the speed of this movement.

Inspired from negative sampling done from the vocabulary, we also try sampling the words from the entire affect lexica instead of using WordNet. This suffers from the semantic relationship problems discussed in step one, the performance, hence declines.

The model used in CBOW is the opposite of the one defined for SkipGram. Given a context window around a word, the model predicts that word. Hence, the hidden layer output is defined as $\mathbf{h} = \frac{1}{C}\sum_{c=1}^{C}\mathbf{v}_{w_c}$, the combination of all input word vectors corresponding to the context of the output word. The loss function is formulated similar to the one already described.

**GloVe (Pennington et al., 2014) Loss Function**: We achieve this by using a weighted regularized version of the original GloVe objective. Unlike Word2Vec, which considers only local co-occurrences, this approach uses global co-occurrences over our entire corpus.

We use the original GloVe implementation[1], first extracting the vocabulary and building a co-occurrence matrix $\mathbf{X}$. Borrowing the notation from Pennington et al. (2014), the objective for GloVe is as follows:

$$J_1 = \sum_{p,q=1}^{V} f(X_{pq})(\mathbf{w}_p^T\tilde{\mathbf{w}}_q + b_p + \tilde{b}_q - logX_{pq})^2, \tag{5}$$

where $V$ is the vocabulary, $b_p$ and $b_q$ are real-valued bias terms, $\mathbf{w}_p$ is the target vector for the word $w_p$ and $\tilde{\mathbf{w}}_q$ is the corresponding context vector for the word $w_q$. $f(X_{pq})$ is a weighting function defined as:

$$f(x) = \begin{cases} (x/x_{max})^\alpha & if \ x < x_{max} \\ 1 & otherwise. \end{cases} \tag{6}$$

Using the same values as in Pennington et al. (2014), we keep $\alpha$ as $\frac{3}{4}$ and $x_{max}$ as 100. With the strength function $S(i,j)$(defined in **Step 2**) in mind, we define a regularization term as follows:

$$J_2 = \frac{1}{2}\sum_{w_i \in L}\sum_{w_j \in L_{pruned}^i} S(i,j)(\mathbf{w}_p - \tilde{\mathbf{w}}_q)^2, \tag{7}$$

---

[1]https://nlp.stanford.edu/projects/glove/

where indices $i$ and $j$ in $L$ correspond to $p$ and $q$ in $V$ respectively. The final objective is obtained as follows:

$$J = J_1 + \lambda J_2, \tag{8}$$

using the same hyperparameter $\lambda$ as before. The obtained expression is similar to the one used in Bollegala et al. (2016), except that instead of using a strength function $S(i, j)$, they use a binary function $R(i, j)$, which indicates whether a relation exists between words $w_i$ and $w_j$ or not. We direct the readers to Bollegala et al. (2016) for the changes in update equations, which also result in similar values.

Figure 2 illustrates the proposed method through the t–distributed Stochastic Neighbor Embedding (t-SNE) visualization of the first two components of a subset of words. The points are colored to identify their positive (blue) or negative (red) valence in the affect lexicon. As shown in Figure 2a, 'accept', 'reject' and 'refuse' were found close together in the original representation. By adding valence information to baseline embeddings (Figure 2b), the model is able to (a) pull words of similar sentiment closer together and (b) pull words of opposite sentiment, further apart. This shows that our method has face–validity in being able to identify and distinguish sentiment polarities within word embeddings.

## 4 EXPERIMENTS AND RESULTS

In this section, we conduct experiments to examine whether incorporating affect information into learning continuous word representations can improve the intrinsic quality of word embeddings and the performance of trained models on downstream tasks. First, we introduce the dataset and then describe the evaluation framework.

### 4.1 DATA

**ukWaC corpus**: We use the ukWaC (Baroni et al., 2009) corpus for building our word embeddings. It is large enough to obtain embeddings of a good quality, while still being tractable in terms of time and space resources. We flattened the dependency-parsed format of this corpus, resulting in 2.2 billion tokens and a vocabulary size of 569,574 words after removing the words having frequency count of less than 20.

We experimented with different values for $\lambda$ and found that the highest similarity score on **RG** word similarity dataset (Rubenstein & Goodenough, 1965) was with a $\lambda = 2$. The details of this experiment are provided in the Appendix. Accordingly, we have used this value for $\lambda$ throughout. We use a window size of 10 and keep the word embedding dimensions as 300. For Word2Vec, we use negative sampling of 15 words for optimization. For GloVe, we found that running the model for 5 iterations was sufficient. On a machine with 8 core Intel 3.4GHz processor and 16GB RAM, the Word2Vec skipgram approach takes 15 hours, CBOW takes under 100 minutes while GloVe takes approximately 15 minutes per iteration.

We compare the performance of the word2vec skipgram, word2vec CBOW and GloVe models for the following settings:

- The baseline approach corresponding to setting $\lambda$ as 0, i.e. only training on the unlabeled ukWaC corpus

- With $\lambda = 2$, incorporating either valence, arousal and dominance scores to the original corpus one at a time. We refer to these models as '+V','+A' and '+D' respectively

- Incorporating the mean weight of valence, arousal and dominance to the original corpus. We refer to this approach as '+VAD'.

- **Comparison against the state of the art**: We reimplemented the approach by Bollegala et al. (2016) on our dataset. The authors use a binary indicator function for the incorporation of various relations such as synonyms and antonyms in the training process. In the original paper, this approach, trained on GloVe embeddings, demonstrably improved the state of the art on standard word similarity and analogy prediction. We pick their best performing model which uses synonym pairs and train it on our ukWaC corpus with the same parameter settings.

- **Comparison against post-training baseline (Append)**: We compare our results against a simple post-training baseline in which we concatenate the word-vectors ($D$-dimensional WV) with VAD-vectors (3-dimensional AV) resulting in a $D + 3$ dimensional vector. The considered dataset might not contain VAD ratings for stop words and proper nouns. Since they are neutral in sentiment (V), strength (D) and arousal (A), for these out-of-dictionary words the VAD vector is set to neutral $\vec{\nu} = [5, 5, 5]$ in all our experiments.

    1. Normalize both embeddings with their L2-Norms (Equation 9). This reduces the individual vectors to unit-length.

$$
\begin{aligned}
x_i &= \frac{x_i}{\sqrt{\sum_{k=1}^{D} x_k^2}} \quad \forall x_i \in WV \\
a_i &= \frac{a_i}{\sqrt{\sum_{k=1}^{3} a_k^2}} \quad \forall a_i \in AV
\end{aligned}
\tag{9}
$$

    2. These regularized distributional word-embeddings are then concatenated with regularized affect scores as shown in Equation 10.

$$ASWV(w) = WV(w) \oplus AV(w) \tag{10}$$

    3. We also perform the standardization process on the resultant $D + 3$ dimensional embeddings. This transforms each dimension in the vector to have unit variance and zero mean so that each dimension contributes approximately proportionately.

$$y_i = \frac{y_i - \mu}{\sigma} \quad \forall y_i \in ASWV \tag{11}$$

    where $\mu$ and $\sigma$ represent the mean and std. deviation respectively.

    The resultant vectors capture affect properties of the word in a distributional setting.

**Warriner affect word list**: We use affective information from Warriner's affect lexicon (Warriner et al., 2013) which comprises 13,915 words tagged on Valence, Arousal and Dominance scores on a scale of 1–9. Valence is the unhappy–happy scale, Arousal is the calm–excited scale and Dominance indicates the forcefulness of expressed affect. In all our experiments, we use a normalized scale from -4 to 4 to take the signed information into account, similar to Sedoc et al. (2017).

## 4.2 EVALUATION FRAMEWORK

We evaluate the proposed method on three standard tasks: predicting the similarity of words on seven benchmark datasets, detecting outliers in semantic clusters on the 8-8-8 dataset (Camacho-Collados & Navigli, 2016) and predicting sentiment on the Stanford Sentiment Treebank (Socher et al., 2013). We then introduce a new dataset and task for formality, frustration and politeness detection for a labeled email corpus.

### 4.2.1 WORD SIMILARITY PREDICTION

The task is to predict the similarity between given two words. We compute the cosine similarity between the corresponding word embeddings of the two words and assign it as the similarity score. We consider seven benchmark datasets:

- **SIMLEX**: the 999 word pairs list (Hill et al., 2016)
- **MC**: 30 word pairs in Miller Charles (Miller & Charles, 1991)
- **MEN**: 3000 pairs of words (Bruni et al., 2012)
- **RG**: 65 word pairs by Rubenstein-Goodenough (Rubenstein & Goodenough, 1965)
- **RW**: 2034 pairs in the Rare-Words dataset (Luong et al., 2013),
- **SCWS**: 2023 word pairs in Stanford's contextual word similarities (Huang et al., 2012)
- **WordSim**: 353 word-pairs in the WordSim test collection (Finkelstein et al., 2001).

Table 1: Performance of proposed approaches on word similarity and outlier detection: **(a)** In terms of Spearman correlation coefficients for the word similarity task on 7 benchmark datasets, **(b)** In terms of the Outlier Position Percentage (OPP) and accuracy scores for the outlier detection task on 8-8-8 dataset, The baseline model refers to the corpus only approach, with $\lambda = 0$. $\lambda$ is set to 2 for all other approaches: using Valence list(+V), Arousal(+A), Dominance(+D) and average score of valence, arousal and dominance (+VAD). **(c)** and **(d)** provides comparison with prior work.

(a)

| Model | Word Similarity | | | | | | |
|---|---|---|---|---|---|---|---|
| | SIMLEX | MC | MEN | RG | RW | SCWS | WordSim |
| **Word2vec CBOW** | | | | | | | |
| baseline | 0.452 | 0.757 | 0.757 | 0.824 | **0.442** | 0.64 | 0.732 |
| +V | **0.459** | 0.791 | 0.756 | 0.827 | 0.438 | 0.641 | 0.733 |
| +A | 0.453 | **0.794** | 0.755 | 0.839 | 0.44 | **0.645** | **0.743** |
| +D | 0.454 | 0.767 | 0.756 | 0.837 | **0.442** | 0.644 | 0.736 |
| +VAD | 0.454 | 0.784 | **0.759** | **0.841** | 0.433 | 0.643 | 0.727 |
| **Word2vec Skipgram** | | | | | | | |
| baseline | **0.383** | **0.721** | **0.749** | 0.74 | **0.43** | **0.643** | **0.728** |
| +V | 0.381 | **0.721** | 0.7 | **0.761** | 0.39 | 0.579 | 0.692 |
| +A | 0.312 | 0.703 | 0.698 | 0.647 | 0.399 | 0.574 | 0.658 |
| +D | 0.356 | 0.707 | 0.689 | 0.723 | 0.393 | 0.575 | 0.689 |
| +VAD | 0.374 | 0.687 | 0.711 | 0.718 | 0.411 | 0.597 | 0.694 |
| **GloVe** | | | | | | | |
| baseline | 0.349 | 0.676 | 0.613 | 0.775 | 0.298 | **0.505** | 0.58 |
| +V | **0.354** | 0.734 | 0.614 | **0.823** | **0.3** | **0.505** | 0.583 |
| +A | 0.348 | 0.703 | 0.612 | 0.778 | 0.291 | 0.502 | 0.576 |
| +D | 0.353 | 0.721 | **0.615** | 0.817 | 0.297 | 0.501 | 0.581 |
| +VAD | 0.349 | **0.736** | 0.613 | 0.812 | 0.297 | 0.502 | **0.584** |
| Append | **0.369** | 0.682 | 0.534 | 0.723 | 0.239 | 0.424 | 0.399 |

(b)

| Model | Outlier Detection | |
|---|---|---|
| | OPP | Accuracy |
| **Word2vec CBOW** | | |
| baseline | 76.172 | 12.5 |
| +V | 77.539 | 15.625 |
| +A | **78.711** | **18.75** |
| +D | 78.125 | 17.188 |
| +VAD | 78.516 | **18.75** |
| **Word2vec Skipgram** | | |
| baseline | **77.93** | **18.75** |
| +V | 76.172 | 17.187 |
| +A | 76.953 | 15.625 |
| +D | 75.586 | 14.062 |
| +VAD | 76.562 | 14.062 |
| **GloVe** | | |
| baseline | 76.172 | 25.0 |
| +V | 75.781 | 26.562 |
| +A | 75.781 | **28.125** |
| +D | **76.367** | **28.125** |
| +VAD | 75.976 | **28.125** |

(c)

| Model | Word Similarity | |
|---|---|---|
| | MC | RG |
| **GloVe** | | |
| baseline | 0.676 | 0.775 |
| +V | 0.734 | **0.823** |
| +A | 0.703 | 0.778 |
| +D | 0.721 | 0.817 |
| +VAD | **0.736** | 0.812 |
| Append | 0.682 | 0.723 |
| Bollegala et al. (2016) | 0.703 | 0.812 |

(d)

| Model | Outlier Detection | |
|---|---|---|
| | OPP | Accuracy |
| **GloVe** | | |
| baseline | 76.172 | 25.0 |
| +V | 75.781 | 26.562 |
| +A | 75.781 | **28.125** |
| +D | **76.367** | **28.125** |
| +VAD | 75.976 | **28.125** |
| Append | 66.992 | 14.063 |
| Bollegala et al. (2016) | 75.781 | 26.562 |

Each word pair in these benchmarks has a manually assigned score which we consider as gold standard ratings. We report the Spearman Correlation Coefficient between the gold standard similarity ratings and the cosine similarity scores assigned by our model.

The results are provided in Table 1. The rows correspond to the approaches, with baselines corresponding to $\lambda = 0$. The columns correspond to the various datasets discussed above. Our proposed modifications show reasonable improvements for GloVe and word2vec CBOW embeddings. The poorest performance was observed for the word2vec skipgram approach, which did not outperform the baseline approach for most of the datasets.

In Table 1 we show that our approach showed modest improvements over the state of the art method by Bollegala et al. (2016) on word similarity for two datasets. We surmise that the nature of the word similarity task and the absence of "opposite" words in the benchmark datasets could be the reason why non-affective approaches are hard to beat for predicting word similarity. For outlier prediction, our approaches gives a performance improvement of 5% over the approach by Bollegala et al. (2016).

### 4.2.2 OUTLIER DETECTION

Word similarity tasks have been widely used for measuring the semantic coherence of vector space models. However, such tasks often suffer from low inter-annotator agreement scores of gold standard datasets. Hence, we also report our results on an outlier detection task (Camacho-Collados & Navigli, 2016), in order to test the quality of semantic clusters in the vector space models. These results are on the 8-8-8 outlier detection dataset (Camacho-Collados & Navigli, 2016) containing 8 different topics, each made up of a cluster of 8 words and 8 possible outliers. Table 1 summa-

rizes the Outlier Position Percentage(OPP score) and Accuracy of our different models. We refer Camacho-Collados & Navigli (2016) to the readers for further details on the dataset and evaluation metrics.

Apart from Word2Vec SkipGram models, our approaches perform well on both OPP and Accuracy scores(higher is better). We observe that the incorporation of affect information in terms of Valence, Arousal and Dominance scores, shows improvements in an Outlier Detection task on an unrelated dataset with topics such as Football Teams, Solar Systems and Car Manufacturers.

Next, we evaluate our approach on two tasks for sentiment detection. Since our approach enriches word embeddings with affect information, we anticipate that our models would outperform the state of the art on these tasks.

### 4.2.3 SENTIMENT PREDICTION

For the sentiment prediction task, we make use of an available Deep Averaging Network[2](DAN) model (Iyyer et al., 2015) with its default parameter settings along with our modified word embeddings.

We report the results on Stanford Sentiment Treebank(SST) dataset (Socher et al., 2013) which contains fine grained sentiment labels for 215,154 phrases in the parse trees of 11,855 sentences. We use the standard splits of the datasets.

Table 2 summarizes the results for sentiment prediction. We report the accuracy values for both fine and binary classification settings. For the latter, we remove all the instances with neutral labels. Our approaches show significant improvements over the baselines for both these metrics.

As evident from Table 2, our approaches perform better on sentiment prediction task than the state of the art approach (Bollegala et al., 2016). We attribute this improvement to our strength function which not only specifies the existence of a relation between two words, but also provides the signed strength associated with it.

### 4.2.4 PREDICTING FORMALITY, FRUSTRATION AND POLITENESS IN EMAILS

Finally, we evaluate the quality our embeddings on an affect prediction task. We use a new dataset consisting of 980 emails from the ENRON dataset (Cohen, 2009), tagged with formality, frustration and politeness scores by human annotators. We use a CNN based regression model which takes our obtained embeddings as inputs and predicts formality, frustration and politeness in emails.

We use a model which stacks a convolutional layer(5 filters of 10X5 size), a pooling layer(5X5 size with stride 5) and a dense layer(size 50, dropout 0.2). Rectified Linear Unit(ReLU) activation is used throughout with Stochastic Gradient Descent(SGD) optimizer and a mean squared loss.

Results are provided in Table 3. For the affect prediction task, we report the mean and standard deviation of Mean Square Error(MSE) over five different train-test splits(test–ratio:0.2) of the dataset corresponding to Formality, Frustration and Politeness. In this case, along with GloVe and CBOW approaches, our modifications over SkipGram baseline (corpus only) show improvements with low standard deviations in MSE values.

## 5 DISCUSSION

We find reasonable improvements by our proposed approaches in all the task-based evaluations. SkipGram based methods perform poorly in word similarity prediction and outlier detection, but do well on sentiment and affect prediction. This difference in performance on downstream tasks, has been discussed before in Faruqui et al. (2016) and Camacho-Collados & Navigli (2016), who point out various issues with word similarity based evaluations such as task subjectivity, low inter annotator agreements and low correlations between the performance of word vectors on word similarity and NLP tasks like text classification, parsing and sentiment analysis. Performance differences can also be attributed to corpus size, which are examined in the Appendix section.

---

[2]https://github.com/miyyer/dan

Table 2: Performance of proposed approaches on sentiment prediction task: **(a)** In terms of Fine and Binary classification accuracies, using the Deep Averaging Network (DAN) model on the Stanford Sentiment Treebank, **(b)** Comparison with prior work. The baseline model refers to the corpus only approach, with $\lambda = 0$. $\lambda$ is set to 2 for all other approaches: using Valence list(+V), Arousal(+A), Dominance(+D) and average strength(+VAD).

(a)

| Model | Sentiment Prediction | |
|---|---|---|
| | Fine | Binary |
| **Word2vec CBOW** | | |
| baseline | **43.53** | 73.97 |
| +V | 43.03 | 74.90 |
| +A | 42.89 | **78.58** |
| +D | 42.99 | 77.59 |
| +VAD | 42.22 | 76.50 |
| **Word2vec Skipgram** | | |
| baseline | 42.89 | 79.79 |
| +V | **43.12** | 79.13 |
| +A | 41.49 | 79.41 |
| +D | 40.23 | **79.85** |
| +VAD | 40.50 | 75.34 |
| **GloVe** | | |
| baseline | 41.40 | 80.07 |
| +V | 41.45 | 78.09 |
| +A | 39.32 | 79.47 |
| +D | **43.26** | 79.21 |
| +VAD | 41.45 | **80.35** |

(b)

| Model | Sentiment Prediction | |
|---|---|---|
| | Fine | Binary |
| **GloVe** | | |
| baseline | 41.40 | 80.07 |
| +V | 41.45 | 78.09 |
| +A | 39.32 | 79.47 |
| +D | **43.26** | 79.21 |
| +VAD | 41.45 | 80.35 |
| Append | 43.10 | **82.76** |
| Bollegala et al. (2016) | 42.26 | 80.42 |

Table 3: Performance of proposed approaches on affect prediction task: **(a)** In terms of Mean Square Error (MSE) values for affect prediction on a labeled email corpus, **(b)** Comparison with prior work. The baseline model refers to the corpus only approach, with $\lambda = 0$. $\lambda$ is set to 2 for all other approaches: using Valence list(+V), Arousal(+A), Dominance(+D) and average strength(+VAD).

(a)

| Model | Affect Prediction in emails(all values X $10^{-2}$) | | | | | |
|---|---|---|---|---|---|---|
| | Formality | | Frustration | | Politeness | |
| | Mean | Std. Dev. | Mean | Std. Dev. | Mean | Std. Dev. |
| **Word2vec CBOW** | | | | | | |
| baseline | 2.93 | 0.35 | 2.40 | 0.34 | 3.09 | 0.14 |
| +V | 2.89 | 0.35 | 2.61 | 0.36 | 3.24 | 0.15 |
| +A | 3.19 | 0.14 | 2.69 | 0.55 | 3.22 | 0.29 |
| +D | **2.82** | 0.22 | **2.12** | 0.27 | **3.04** | 0.36 |
| +VAD | 2.94 | 0.14 | 2.33 | 0.28 | 3.28 | 0.34 |
| **Word2vec Skipgram** | | | | | | |
| baseline | 2.80 | 0.29 | 2.37 | 0.52 | 3.26 | 0.38 |
| +V | 3.18 | 0.33 | **2.28** | 0.25 | 3.26 | 0.25 |
| +A | **2.79** | 0.36 | 2.42 | 0.23 | 3.22 | 0.28 |
| +D | 3.00 | 0.34 | 2.85 | 0.38 | **3.10** | 0.16 |
| +VAD | 3.26 | 0.43 | 2.62 | 0.40 | 3.25 | 0.29 |
| **GloVe** | | | | | | |
| baseline | 3.12 | 0.17 | 2.59 | 0.35 | 3.26 | 0.22 |
| +V | **2.93** | 0.28 | **2.32** | 0.37 | 3.19 | 0.11 |
| +A | **2.93** | 0.43 | 2.53 | 0.25 | **3.13** | 0.25 |
| +D | 3.11 | 0.36 | 2.50 | 0.42 | 3.27 | 0.25 |
| +VAD | 3.47 | 0.26 | 2.51 | 0.30 | 3.49 | 0.38 |

(b)

| Model | Affect Prediction in emails | | | | | |
|---|---|---|---|---|---|---|
| | Formality | | Frustration | | Politeness | |
| | Mean | Std. Dev. | Mean | Std. Dev. | Mean | Std. Dev. |
| **GloVe** | | | | | | |
| baseline | 3.12 | 0.17 | 2.59 | 0.35 | 3.26 | 0.22 |
| +V | 2.93 | 0.28 | **2.32** | 0.37 | 3.19 | 0.11 |
| +A | 2.93 | 0.43 | 2.53 | 0.25 | **3.13** | 0.25 |
| +D | 3.11 | 0.36 | 2.50 | 0.42 | 3.27 | 0.25 |
| +VAD | 3.47 | 0.26 | 2.51 | 0.30 | 3.49 | 0.38 |
| Append | **2.89** | 0.18 | 2.34 | 0.44 | 3.28 | 0.30 |
| Bollegala et al. (2016) | 3.03 | 0.30 | 2.59 | 0.48 | 3.37 | 0.16 |

The results suggest that different embeddings perform well for different tasks. In word similarity tasks, the **+V** model performs well in GloVe setting but the **+A** model seems to perform the best for CBOW. Similar results are observed in sentiment prediction: for binary sentiment prediction, arousal scores give the best performance with CBOW embeddings but dominance and valence give the best performance with skip-gram and GloVe embeddings respectively. This suggests that the most flexible method could be an ensemble implementation that considers all these inputs before

Table 4: Fine grained predictions for GloVe based models on instances from Stanford Sentiment Treebank(SST) dataset. "baseline" refers to the corpus only approach, +Valence refers to our baseline+Valence affect lexicon model and Bollegala et al. (2016) uses WordNet synonyms along with the corpus. Class labels are from +2(Very positive) to -2(Very Negative).

| Serial No. | Sentence | True label | baseline | +Valence | (Bollegala et al., 2016) |
|---|---|---|---|---|---|
| 1 | if this movie were a book , it would be a page-turner , you ca n't wait to see what happens next . | +2 | 0 | -1 | 0 |
| 2 | the movie will reach far beyond its core demographic . | +2 | -1 | -1 | -1 |
| 3 | offers that rare combination of entertainment and education . | +2 | +2 | +2 | +2 |
| 4 | a smart , sweet and playful romantic comedy . | +2 | +2 | +2 | +2 |
| 5 | a dreadful live-action movie . | -2 | +2 | -2 | +2 |
| 6 | hubac 's script is a gem . | +2 | -1 | +1 | +1 |
| 7 | i hate this movie | -1 | +2 | -2 | +2 |

Table 5: Top 10 nearest neighbors of the word 'dreadful' in terms of cosine similarity for three GloVe based models.

(a) baseline model

| Word | Similarity |
|---|---|
| terrible | 0.853 |
| awful | 0.844 |
| horrible | 0.778 |
| appalling | 0.752 |
| horrendous | 0.725 |
| ghastly | 0.723 |
| atrocious | 0.672 |
| frightful | 0.671 |
| horrid | 0.664 |
| enfant | 0.662 |

(b) baseline+Valence

| Word | Similarity |
|---|---|
| terrible | 0.932 |
| awful | 0.930 |
| horrendous | 0.925 |
| atrocious | 0.909 |
| frightful | 0.903 |
| dread | 0.885 |
| unspeakable | 0.883 |
| fearful | 0.870 |
| abominable | 0.863 |
| horrific | 0.858 |

(c) (Bollegala et al., 2016)

| Word | Similarity |
|---|---|
| terrible | 0.860 |
| awful | 0.848 |
| horrible | 0.792 |
| appalling | 0.750 |
| horrendous | 0.743 |
| ghastly | 0.719 |
| frightful | 0.671 |
| atrocious | 0.671 |
| horrid | 0.669 |
| enfant | 0.662 |

predicting a final class. Also note that given the vocabulary of our ukWaC corpus as $569,574$ words, our affect lexica with $13,915$ words is relatively small. We plan to take this work forward by further analysis in the future. At the least, we expect superior word embeddings with better quality and larger affect lexica.

## 5.1 AFFECT-POLYSEMY

Words may have different affect in different contexts. This happens because their meanings change from one context to another. For example "A wicked woman came in" vs a slang "This game is wicked" should have different affect scores for both "wicked" because the meaning of the word almost turns on its head (from "devilish" to "cool/commendable"). But datasets like Warriner et al. (2013) do not capture this as the scores attributed by human annotators inherently average out the contexts (to the best of their knowledge) in which the words tend to appear. This problem (of "affect-polysemy") is related to "polysemy" which cripples today's word-level embedding models(Gladkova & Drozd (2016), Huang et al. (2012)). Word-level vector embeddings like SkipGram combine multiple senses of a polysemous word into single embedding vector (Liu et al. (2015)). Although, there have been a few attempts to address this shortcoming using multi-prototype vector-space models (Reisinger & Mooney (2010), Huang et al. (2012), Tian et al. (2014)), but they also face several challenges (Liu et al. (2015)).

Since, single vector embeddings are still the most popular choice for NLP tasks, the proposed approach tries to improve the quality of these vanilla embeddings for tasks which can benefit by inducing affect information. Therefore, in this paper, we evaluate our approach on sentence/phrase level prediction tasks like sentiment prediction and tone of emails. The underlying state-of-the-art recurrent/convolutional models try to create context-aware sentence level representations from word-embeddings to predict information like sentiment, formality etc. Our affect induced word-embeddings perform better than vanilla embeddings on these tasks thus supporting our claim.

## 6 ERROR ANALYSIS

We consider three models for analysis: GloVe baseline(corpus only approach), GloVe + Valence and GloVe + synonyms (Bollegala et al., 2016). Table 4 shows the true class label and predictions for a few instances from SST test set. We discuss these results below.

For the first two sentences, although the overall sentiment is clearly positive, all the three the models fail to capture this. The models fail in both fine-grained and binary(positive/negative) settings. The reason for poor predictions can be attributed to the absence of any affect-sensitive words. On the other hand, third and fourth sentences are rich with affect information. Words like 'entertainment', 'smart', 'sweet' and 'playful' make it easier for the models to predict the positive sentiment in these two sentences.

The fifth sentence has a negative overall sentiment, with a true label of -2. Surprisingly, baseline model and Bollegala et al. (2016) approaches fail to capture this. However, our approach is able to make the correct prediction. This observation can be explained as follows. The main affect word in the sentence is "dreadful", which is present in Warriner affect lexicon but not in synonym word pairs used by Bollegala et al. (2016). This enables our model to get improved embedding for this word. This infact can be observed in the neighborhood of 'dreadful' compared in Table 5. The embedding for 'dreadful' is a lot closer to negative words for our approach than others.

We observe a similar behavior in last two sentences. The better quality embeddings learnt for 'gem' and 'hate' help to make better predictions while other models make errors.

Including various other instances which we observe, the presence or absence or affect words in the lexicon plays a vital role. Since we work in a joint learning framework, embeddings for other words are also improved, but this improvement is significant for words present in the lexicon only. The inconsistency in the performance of our models can be attributed to this along with the quality of affect lexicon itself.

## 7 CONCLUSION

This work proposes methods to incorporate information from an affect lexicon into Word2Vec and GloVe training process. In a nutshell, we first use WordNet to identify word pairs in the affect lexicon which are semantically related. We define the strength of this relationship using available affect scores. Finally, we modify the training objectives to incorporate this information. In order to evaluate our embeddings, we compare them with baseline approaches where the training completely ignores the affect information. Our embeddings show improvements over baselines on not only Word Similarity benchmarks but also on a more complex, Outlier Detection task. We also do this comparison extrinsically and show that our modified embeddings perform better over prior work in predicting sentiment and predicting formality, frustration and politeness in emails. Among models using Valence, Arousal or Dominance score lists, there is no clear winner but overall addition of valence scores does a reasonable job in almost all of the cases.

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

# Appendix

1. **Choosing an appropriate value for hyper-parameter** $\lambda$:

   In order to choose a suitable value for $\lambda$, we take a 100 MB sample of ukWaC corpus. The sample has close to 20 million tokens, with a vocabulary size of 27,978 words, eliminating all the words having the frequency count of less than 20. We choose a smaller corpus for tuning as it is more manageable with respect to space and time resources.

   We train a Word2Vec SkipGram model on the above 100MB sample and Valence affect lists by using all the $\lambda$ value from the set $(0, 0.5, 1, 2, 10, 100, 1000)$ one by one.

   To pick the most suitable value, we compare the results on word similarity task on the Rubenstein-Goodenough(**RG**) dataset (Rubenstein & Goodenough, 1965). The results are given in Figure 3. Since $\lambda = 2.0$ performs the best, we fix this value for all our experiments.

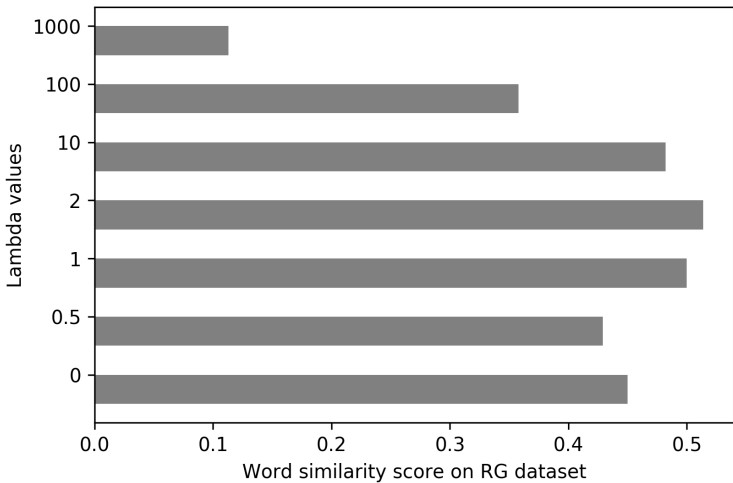

Figure 3: Word similarity scores(Spearman correlation coefficient) on RG dataset for a Word2Vec SkipGram model trained on a 100 MB sample of ukWaC corpus and Valence affect scores using different $\lambda$ values.

2. **Studying the effect of corpus size**:

   We conduct an error analysis of the poor performance of Word2Vec Skipgram by observing the effect of varying the corpus size. We take different sized samples from the ukWaC corpus and report the word similarity performance on RG dataset in Figure 4.

   We observe irregularities in the performance of baseline approach($\lambda$=0). Adding the affective information has a negative impact for corpora with sizes 2.5GB and 4.5GB while shows minor improvements over baseline for larger corpora. This improvement over baseline is the most for a smaller, 100MB corpus. We believe better preprocessing on the corpus should help with these non-intuitive observations.

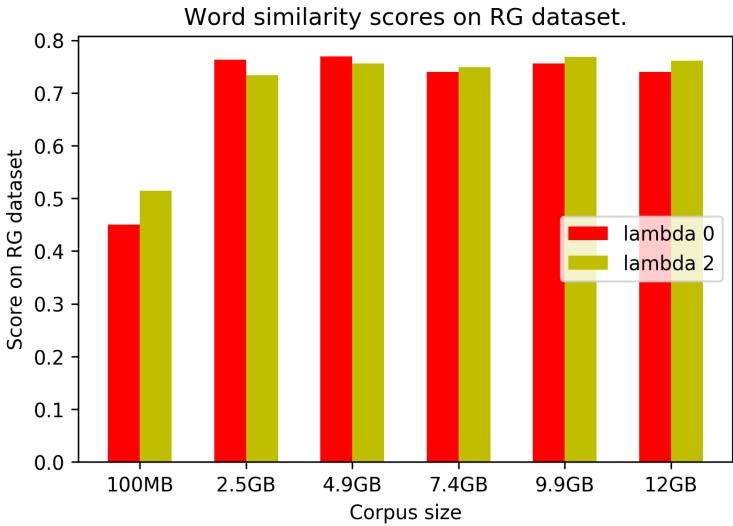

Figure 4: Word similarity scores(Spearman correlation coefficient) on RG dataset for a Word2Vec SkipGram model trained on corpus with different sizes using $\lambda$ values as 0 and 2 with valence affect list.

