# OpenReview forum: "Towards Building Affect sensitive Word Distributions"
_ICLR.cc/2018/Conference — Reject_

### Official Review · AnonReviewer3 · 2017-11-27
**Interesting paper with promising results**

**Rating:** 6
**Confidence:** 3

**Review:**

This paper proposed to use affect lexica to improve word embeddings. They extended the training objective functions of Word2vec and Glove with the affect information. The resulting embeddings were evaluated not only on word similarity tasks but also on a bunch of downstream applications such as sentiment analysis. Their experimental results showed that their proposed embeddings outperformed standard Word2vec and Glove. In sum, it is an interesting paper with promising results and the proposed methods were carefully evaluated in many setups.

Some detailed comments are:
-	Although the use of affect lexica is innovative, the idea of extending the training objective function with lexica information is not new. Almost the same method was proposed in K.A. Nguyen, S. Schulte im Walde, N.T. Vu. Integrating Distributional Lexical Contrast into Word Embeddings for Antonym-Synonym Distinction. In Proceedings of ACL, 2016.
-	Although the lexicons for valence, arousal, and dominance provide different information, their combination did not perform best. Do the authors have any intuition why?
-	In Figure 2, the authors picked four words to show that valence is helpful to improve Glove word beddings. It is not convincing enough for me.  I would like to see to the top k nearest neighbors of each of those words.

---

> ### Author Response · Authors · 2018-01-05
> **Performed Error Analysis to better understand our results and provided additional information as requested**
>
> Thank you for a detailed and a lucid review. We would like to address the points made individually:
>
> 1) Detailed comment 1 - "Although the use of affect lexica is innovative...": Using the affect lexicon is different than prior work due to the presence of word level scores on a continuous scale instead of discrete labels. Hence, it demands for a novel way to define the strength between two words. This is covered by our Algorithm 1 in the paper. Our intuition for joint learning comes from how Word2Vec and GloVe training models work. Since this has been used to incorporate knowledge bases in the prior work, we end up with a similar looking loss term. However, we refer you to our Related Work section (see Section 2), where we individually point out the similarities and differences with the prior work in terms of the intuition and the mathematical formulations.
>
> 2) Detailed comment 2 - "combination of valence, arousal and dominance did not perform best": The quality of embeddings itself can partially be the reason why the combination of all the affect scores does not perform the best in all cases. Exploring better techniques for combining the information is a part of future exploration.
>
> In order to further understand our results, we also perform error analysis for sentiment prediction task. We have added our observations in a separate section (see Section 6) in the paper.
>
> 3) Detailed comment 3 - "top k nearest neighbors":
>
> Here, we show the top 5 neighbors of each of the four words shown in Figure 2. We also show corresponding cosine similarity values.
>
> i) Refuse-
>
> GloVe baseline	|	GloVe + Valence information
>
> insist, 0.716           |	deny, 0.721
> refusal, 0.707	|	reject, 0.707
> decide, 0.682	        |	insist, 0.706
> deny, 0.672		|	refusal, 0.697
> reject, 0.671	        |	decide, 0.680
>
> ii) Reject-
>
> GloVe baseline	|	GloVe + Valence information
>
> accept, 0.708	        |	accept, 0.726
> refuse, 0.671	        |	refuse, 0.707
> oppose, 0.659	|	oppose, 0.698
> dismiss, 0.657	|	dismiss, 0.692
> deny, 0.657		|	deny, 0.691
>
> iii) Accept-
>
> GloVe baseline					|	GloVe + Valence information
>
> cannot, 0.727					|	reject, 0.726
> reject, 0.708					        |	cannot, 0.708
> Visa/Mastercard/Switch, 0.697	        | 	acknowledge, 0.701
> acknowledge, 0.696				|	Visa/Mastercard/Switch, 0.688
> must, 0.671						|	Dogs/pets, 0.676
>
> iv) Approve-
>
> GloVe baseline				|	GloVe + Valence information
>
> approval, 0.740				|	approval, 0.731
> agree, 0.669				        |	endorse, 0.679
> a.fldnoofproducts, 0.648	        |	agree, 0.657
> proposal, 0.646				|	a.fldnoofproducts, 0.646
> endorse, 0.644				|	proposal, 0.625
>
>  In general, among the cases which we analyze, we observe that using the affect information takes a word closer to the synonyms and farther from antonyms. However, there are exceptions to this. For instance, the similarity between 'reject' and 'accept' increases after the addition of valence information. We again refer you to the error analysis section where we describe various other instances and point out several possible reasons for errors.

---

### Official Review · AnonReviewer1 · 2017-11-28
**A curious example of adding structured knowledge into embedding spaces**

**Rating:** 4
**Confidence:** 5

**Review:**

This paper proposes integrating information from a semantic resource that quantifies the affect of different words into a text-based word embedding algorithm.

The affect lexical seems to be a very interesting resource (although I'm not sure what it means to call it 'state of the art'), and definitely support the endeavour to make language models more reflective of complex semantic and pragmatic phenomena such as affect and sentiment.

The justification for why we might want to do this with word embeddings in the manner proposed seems a little unconvincing to me:

- The statement that 'delighted' and 'disappointed' will have similar contexts is not evident to me at least (other then them both being participle / adjectives).

- Affect in language seems to me to be a very contextual phenomenon. Only a tiny subset of words have intrinsic and context-free affect. Most affect seems to me to come from the use of words in (phrasal, and extra-linguistic) contexts, so a more context-dependent model, in which affect is computed over phrases or sentences, would seem to be more appropriate. Consider words like 'expensive', 'wicked', 'elimination'...

The model proposes several applications (sentiment prediction, predicting email tone, word similarity) where the affect-based embeddings yield small improvements. However, in different cases, taking different flavours of affect information (V, A or D) produces the best score, so it is not clear what to conclude about what sort of information is most useful.

It is not surprising to me that an algorithm that uses both WordNet and running text to compute word similarity scores improves over one that uses just running text. It also not surprising that adding information about affect improves the ability to predict sentiment and the tone of emails.

To understand the importance of the proposed algorithm (rather than just the addition of additional data), I would like to see comparison with various different post-processing techniques using WordNet and the affect lexicon (i.e. not just Bollelaga et al.) including some much simpler baselines. For instance, what about averaging WordNet path-based distance metrics and distance in word embedding space (for word similarity), and other ways of applying the affect data to email tone prediction?

---

> ### Author Response · Authors · 2018-01-05
> **Added clarifications, performed error analysis and conducted additional experiments as requested**
>
> Thank you for a detailed and a lucid review. We would like to address the points made individually:
>
> 1) Explanation for calling "state-of-the-art": We apologise if the language is unclear at some point in the paper. But we do not mean to refer the affect lexicon itself as 'state-of-the-art'. We use 'state-of-the-art' for [3] which uses synonym pairs to modify GloVe embeddings. To have a proper comparison, we run this approach on our dataset and compare it to our results for all the evaluation tasks. I refer you to the 'Experiments and Results' section (Section 4) in the paper for more details.
>
> 2) With reference to the 'delighted' and 'disappointed' example, we were citing the work by [1] who applied post-hoc affect-based signed clustering on word embeddings and identified that after incorporating valence ratings using the signed clustering algorithm, 'disappointed' moves further away from 'delighted' than in the original space. More details are available in the original paper.
>
> 3) "Affect as a contextual phenomenon": We agree with the observation that words may have different affect in different contexts. We refer you to Section 5.1 in the paper on "Affect Polysemy", which we have added to discuss this issue in detail.
>
> 4) "Small improvements, not clear as to what to conclude": In order to test the significance of our improvements, we perform hypothesis testing. Taking advice from prior work, we use Fisher transformation to test for statistical significance of our word similarity correlations. We were able to achieve a similar level of significance as prior work such as [2] and [3].
>
> Apart from this, we have added an error analysis section (see Section 6) in the paper to discuss reasons for inconsistent performance and why models make errors. We mainly focus on sentiment prediction task. We qualitative compare our approach with [3] and baseline approaches. Overall, we see a reasonable improvement in almost all the cases with the addition of valence affect scores.
>
> 5) Comparison with Post processing techniques: To build a stronger evaluation, taking this advice, we have added comparison with a post-processing baseline in the paper. We refer you to Section 4.1 where we explain this approach. We have added the corresponding results in the Evaluation Framework (section 4.2).
>
> References:
>
> [1] Sedoc, J., et al. "Predicting Emotional Word Ratings using Distributional Representations and Signed Clustering". Proceedings of the European Association for Computational Linguistics. ACL, 2017.
>
> [2] Xu, Chang, et al. "Rc-net: A general framework for incorporating knowledge into word representations." Proceedings of the 23rd ACM International Conference on Conference on Information and Knowledge Management. ACM, 2014.
>
> [3] Bollegala, Danushka, et al. "Joint Word Representation Learning Using a Corpus and a Semantic Lexicon." AAAI. 2016.

---

### Official Review · AnonReviewer2 · 2017-11-30
**Lack of novelty and unconvincing results**

**Rating:** 4
**Confidence:** 4

**Review:**

This paper introduces modifications the word2vec and GloVe loss functions to incorporate affect lexica to facilitate the learning of affect-sensitive word embeddings. The resulting word embeddings are evaluated on a number of standard tasks including word similarity, outlier prediction, sentiment detection, and also on a new task for formality, frustration, and politeness detection.

A considerable amount of prior work has investigated reformulating unsupervised word embedding objectives to incorporate external resources for improving representation learning. The methodologies of Kiela et al (2015) and Bollegala et al (2016) are very similar to those proposed in this work. The main originality seems to be captured in Algorithm 1, which computes the strength between two words. Unlike prior work, this is a real-valued instead of a binary quantity. Because this modification is not particularly novel, I believe this paper should primarily be judged based upon the effectiveness of the method rather than the specifics of the underlying techniques. In this light, the performance relative to the baselines is particularly important. From the results reported in Tables 1, 2, and 3, I do not see compelling evidence that +V, +A, +D, or +VAD consistently lead to significant performance increases relative to the baseline methods. I therefore cannot recommend this paper for publication.

---

> ### Author Response · Authors · 2018-01-05
> **Performed Hypothesis Testing to test the significance of our improvements and Error Analysis to analyse our results**
>
> Thank you for a detailed and a lucid review. We would like to address the points made individually:
>
> 1) Lack of novelty:
>
>  To the best of our knowledge, there is no prior art which incorporates affect lexicons in a joint learning framework with either Word2Vec or GloVe. Having word level scores on a continuous scale instead of discrete labels in the lexicon seems more useful but it demands for a novel way to define the strength between two words. This is covered by our Algorithm 1 in the paper. For incorporating this information in a joint learning framework, we take our intuition from how Word2Vec and GloVe models work. Since this has been used to incorporate knowledge bases in the prior work, we end up with a similar looking loss term. However, we refer you to our Related Work section (see Section 2), where we individually point out the similarities and differences with the prior work in terms of the intuition and the mathematical formulations.
>
> 2) Unconvincing results:
>
> To justify the significance of our improvements, we perform a hypothesis test. Taking advice from prior work, we use Fisher transformation to test for statistical significance of our word similarity correlations. We were able to achieve a similar level of significance as prior work such as [1] and [2].
>
> To partially address the inconsistent performance of our models across various tasks and analyze other reasons for errors, we perform an error analysis on sentiment detection task. We refer you to the Error Analysis section (Section 6) in the paper which we have added to discuss our observations.
>
> References:
>
> [1] Xu, Chang, et al. "Rc-net: A general framework for incorporating knowledge into word representations." Proceedings of the 23rd ACM International Conference on Conference on Information and Knowledge Management. ACM, 2014.
>
> [2] Bollegala, Danushka, et al. "Joint Word Representation Learning Using a Corpus and a Semantic Lexicon." AAAI. 2016.

---

### Author Response · Authors · 2018-01-05
**Summarising the major additions done in the paper**

The following changes have been made to the paper based on the Reviewer Comments.

1) Comparison against post-training baseline: The paper presents a pretraining based approach to create enriched word embeddings. In order to have a complete evaluation, we compare our results to a post-training method.

Once we have trained embeddings using a standard approach, we modify that embedding space in a post-processing step to inculcate the affect information. The detailed explanation of this approach in added to Section 4.1. The results have been added in the Evaluation Framework (section 4.2).

2) Affect Polysemy: We agree with AnonReviewer1 that words may have different affect in different contexts. A new discussion on this has been added to Section 5.1.

3) Error Analysis: In order to gain further insights into our results, we perform an error analysis for sentiment prediction task. We qualitatively compare our approach with [1] and baseline methods. We have added our observations as a new Error Analysis section (see Section 6).

References:

[1] Bollegala, Danushka, et al. "Joint Word Representation Learning Using a Corpus and a Semantic Lexicon." AAAI. 2016.

---

### Decision · Program_Chairs · 2018-01-29
**ICLR 2018 Conference Acceptance Decision**

**Decision:**

Reject

**Comment:**

This work attempts to incorporate affect information from additional resources into word embeddings. This is a valuable goal, but the methods used are very similar to existing ones, and the experimental results are not quite convincing enough to make a strong enough case for accepting the paper.